# Cellular Prion Protein Expression in the Brain Tissue from *Brucella ceti*-Infected Striped Dolphins (*Stenella coeruleoalba*)

**DOI:** 10.3390/ani12101304

**Published:** 2022-05-19

**Authors:** Clotilde Beatrice Angelucci, Roberto Giacominelli-Stuffler, Marina Baffoni, Cristina Esmeralda Di Francesco, Gabriella Di Francesco, Ludovica Di Renzo, Manuela Tittarelli, Antonio Petrella, Carla Grattarola, Sandro Mazzariol, Eva Sierra, Antonio Fernández, Giovanni Di Guardo

**Affiliations:** 1Faculty of Veterinary Medicine, University of Teramo, 64100 Teramo, Italy; bcangelucci@unite.it (C.B.A.); rgiacominellistuffler@unite.it (R.G.-S.); mbaffoni@unite.it (M.B.); cedifrancesco@unite.it (C.E.D.F.); 2Istituto Zooprofilattico Sperimentale (IZS) dell’Abruzzo e Molise “G. Caporale”, 64100 Teramo, Italy; g.difrancesco@izs.it (G.D.F.); l.direnzo@izs.it (L.D.R.); m.tittarelli@izs.it (M.T.); 3IZS della Puglia e Basilicata, 71121 Foggia, Italy; antonio.petrella@izspb.it; 4IZS del Piemonte, Liguria e Valle d’Aosta, CReDiMa, 10154 Turin, Italy; carla.grattarola@izsto.it; 5Department of Comparative Biomedicine and Food Science, University of Padua, 35020 Legnaro, Italy; sandro.mazzariol@unipd.it; 6Faculty of Veterinary Medicine, University of Las Palmas de Gran Canaria, Las Palmas de Gran Canaria, 35416 Canary Islands, Spain; eva.sierra@ulpgc.es (E.S.); antonio.fernandez@ulpgc.es (A.F.)

**Keywords:** *Brucella ceti*, cellular prion protein, striped dolphin, *Stenella coeruloeoalba*, brain, neurobrucellosis, pathogenesis, infection

## Abstract

**Simple Summary:**

*Brucella ceti*, a zoonotic bacterial pathogen, is known to exhibit a strong neurotropism and neuropathogenicity for striped dolphins (*Stenella coeruleoalba*), often leading to their stranding and death. Given the lack of information on *B. ceti* infection’s neuropathogenesis, we investigated, for the first time, cellular prion protein (PrP^c^) expression in the brain tissue from *B. ceti*-infected, neurobrucellosis-affected striped dolphins. Our study was inspired by previous work, reporting PrP^c^ as the host cell receptor for *B. abortus* on the surface of murine macrophages. Immunohistochemistry (IHC) and Western blot (WB) analyses were carried out on brain tissues from 12 striped dolphins found stranded along the coasts of Italy (11 specimens) and the Canary Islands (one individual), five of which served as negative controls. While PrP^c^ IHC yielded inconclusive results, WB analyses showed a clear-cut PrP^c^ expression, albeit of different intensity, in the brain tissue of all the herein investigated, *B. ceti*-infected and neurobrucellosis-affected individuals. In this respect, the aforementioned PrP^c^ expression patterns could be influenced by a number of intrinsic host-related factors, as well as by several extrinsic factors including simultaneously occurring neuropathies and/or coinfections by other neurotropic pathogens. Additionally, an upregulation of PrP^c^ mRNA in the brain tissue of striped dolphins could be also hypothesized during the different stages of *B. ceti* infection, in a similar fashion to what is already shown in murine bone marrow cells challenged with *Escherichia coli*. In conclusion, much more work is needed in order to properly assess the role of PrP^c^, if any, as a host cell receptor for *B. ceti* in striped dolphins.

**Abstract:**

*Brucella ceti*, a zoonotic pathogen of major concern to cetacean health and conservation, is responsible for severe meningo-encephalitic/myelitic lesions in striped dolphins (*Stenella coeruleoalba*), often leading to their stranding and death. This study investigated, for the first time, the cellular prion protein (PrP^c^) expression in the brain tissue from *B. ceti*-infected, neurobrucellosis-affected striped dolphins. Seven *B. ceti*-infected, neurobrucellosis-affected striped dolphins, found stranded along the Italian coastline (6) and in the Canary Islands (1), were investigated, along with five *B. ceti*-uninfected striped dolphins from the coast of Italy, carrying no brain lesions, which served as negative controls. Western Blot (WB) and immunohistochemistry (IHC) with an anti-PrP murine monoclonal antibody were carried out on the brain parenchyma of these dolphins. While PrP^c^ IHC yielded inconclusive results, a clear-cut PrP^c^ expression of different intensity was found by means of WB analyses in the brain tissue of all the seven herein investigated, *B. ceti*-infected and neurobrucellosis-affected cetacean specimens, with two dolphins stranded along the Italian coastline and one dolphin beached in Canary Islands also exhibiting a statistically significant increase in cerebral PrP^c^ expression as compared to the five *Brucella* spp.-negative control specimens. The significantly increased PrP^c^ expression found in three out of seven *B. ceti*-infected, neurobrucellosis-affected striped dolphins does not allow us to draw any firm conclusion(s) about the putative role of PrP^c^ as a host cell receptor for *B. ceti*. Should this be the case, an upregulation of PrP^c^ mRNA in the brain tissue of neurobrucellosis-affected striped dolphins could be hypothesized during the different stages of *B. ceti* infection, as previously shown in murine bone marrow cells challenged with *Escherichia coli*. Noteworthy, the inflammatory infiltrates seen in the brain and in the cervico-thoracic spinal cord segments from the herein investigated, *B. ceti*-infected and neurobrucellosis-affected striped dolphins were densely populated by macrophage/histiocyte cells, often harboring *Brucella* spp. antigen in their cytoplasm, similarly to what was reported in macrophages from mice experimentally challenged with *B. abortus*. Notwithstanding the above, much more work is needed in order to properly assess the role of PrP^c^, if any, as a host cell receptor for *B. ceti* in striped dolphins.

## 1. Introduction

*Brucella* genus members are a group of Gram-negative, pathogenic bacteria causing brucellosis in many mammalian species, including aquatic mammals. 

*Brucella ceti* (a smooth *Brucella* spp. microorganism), which is known to infect dolphins and whales from the two Hemispheres, has been isolated in recent years from striped dolphins (*Stenella coeruleoalba*) stranded along the Italian coastline [1,2,3], with neurobrucellosis being a quite common and peculiar pathological feature in these animals, as previously reported elsewhere [4,5,6].

Prion diseases (PDs), or transmissible spongiform encephalopathies (TSEs), are fatal, progressive neurodegenerative disorders characterized by the accumulation, in the central nervous system (CNS) as well as in peripheral nervous and extraneural tissues, of a pathological isoform, PrP^Sc^, of the host-encoded, cellular prion protein (PrP^c^) [7]. While PD/TSE-associated lesions are limited to the host’s CNS, the disease-specific protein PrP^Sc^ accumulates in secondary lymphoid tissues prior to neuroinvasion, with PrP^Sc^ deposition and prion agent replication depending upon the PrP^c^ expression levels on host cells. Within lymphoid tissues, PrP^c^ may additionally undergo an up- or downregulation during adaptive immune responses [8]. Besides PDs/TSEs, PrP^c^ is pathogenetically involved in other neurological disorders and infectious diseases, including Alzheimer’s disease and human immunodeficiency virus (HIV) infection [8,9]. 

PrP^c^ is a small glycoprotein consisting of 253, 254, 256, and 257 amino acids in man, mink, sheep, and goat, respectively, and is also characterized by a C-helix C-terminal region stabilized by a disulfide bridge; this region ranges from residue 125 to 231; the N-terminal region is very flexible and ranges from residue 23 to 124 [10,11]. Differently from PrP^c^, PrP^Sc^ is misfolded, thereby exhibiting greater resistance to proteinase K (PK) digestion and being highly insoluble in non-ionic detergents. Although its biological functions are still far from being fully elucidated, PrP^c^ is likely involved in the transport of ionic copper (Cu^++^) from the external environment into cells. A role in the formation of neuronal synapses and cellular signaling has been also suggested, with PrP^c^ likely serving as a host cell receptor for a number of microbial pathogens [12,13]. As far as extraneural expression is concerned, phagocytes, dendritic cells (DCs), lymphocytes and natural killer (NK) cells have been shown to harbor PrP^c^ on their surface, with its distribution varying between humans and mice [7]. 

With specific reference to microbial pathogens, a key role played by PrP^c^ in allowing *B. abortus* entry into murine macrophages has been additionally demonstrated following experimental infection [14,15,16].

The present study, which was inspired by a previously published theoretical article [17], was aimed at identifying and evaluating brain PrP^c^ expression in a number of beached striped dolphins, while also attempting to uncover the existence of pathogenetic relationships-if any-between PrP^c^ expression and *Brucella* spp. immunoreactivity (IR) in the cerebral tissue of *B. ceti*-infected, neurobrucellosis-affected striped dolphins, as compared to *B. ceti*-negative striped dolphins with no microscopic evidence of CNS lesions.

## 2. Materials & Methods

A total number of 12 striped dolphins, seven affected by neurobrucellosis due to *B. ceti* and five *B. ceti*-negative and with no microscopic evidence of CNS lesions, were included in this study.

All the aforementioned cetaceans were found stranded lifeless along the coast of Italy, with the only exception of a *B. ceti*-infected, neurobrucellosis-affected individual, that was found stranded in 2004 in Canary Islands. As far as concerns the *post mortem* preservation/conservation code of the herein investigated striped dolphin specimens, this was categorized as “good” (code 2), according to the classical/canonical classification scheme by Geraci and Lounsbury [18].

During *post-mortem* examination, all major organs from each animal-including the brain and the cervico-thoracic spinal cord-were promptly fixed in 10% neutral buffered formalin, to be subsequently embedded in paraffin and cut into 5-micron-thick sections for light microscopic analyses. After its removal from the skull, the brain was cut into two halves, one of which was immediately fixed in 10% neutral buffered formalin, while the remaining one was frozen at −80 °C for microbiological, biomolecular, and immunobiochemical ancillary investigations. More in detail, microbiological analyses were performed on each dolphin specimen, by means of selective media, against bacterial (both aerobic and anaerobic) and fungal pathogens potentially impacting cetacean health and conservation. For *Brucella* spp. isolation, we followed the technique described in the OIE Manual of Diagnostic Tests and Vaccines [19], using both selective and non-selective solid media and enrichment broths. Cultures were attempted from the CNS (brain and spinal cord) as well as from extraneural organs and tissues (spleen, liver, kidney, lung, heart, and lymph nodes). *Brucella* spp. strains recovered from the herein investigated striped dolphins were identified as *B. ceti* using the PCR-RFLP method [20], then subjected to genomic analysis at the National and OIE Reference Laboratory for Brucellosis, Istituto Zooprofilattico Sperimentale dell’Abruzzo e del Molise (IZSAM), Teramo, Italy.

In-depth biomolecular and immunohistochemical (IHC) investigations were also carried out against *Cetacean morbillivirus* (CeMV) and *Toxoplasma gondii*, two pathogens of concern for striped dolphins alongside *B. ceti* [5,21,22,23]. The aforementioned biomolecular (RT-PCR and PCR for CeMV and *T. gondii*, respectively) and IHC analyses were performed following *ad hoc*, previously published laboratory protocols [22,23].

For PrP^c^ detection, we followed a protocol described elsewhere [24], with Western blot (WB) being the technique utilized. Indeed, WB ranks among the “rapid tests” employed in Italy and in the European Union for the active surveillance of ruminant PDs/TSEs [25]. To this aim, the commercially available anti-PrP monoclonal antibody (Mab) F89/160.1.5 (*VMRD*, Inc., Pullman, WA, USA) was used, with a four amino acid-bearing (142–145) PrP epitope (IHFG) being specifically recognized by F89/160.1.5. The same MAb was additionally employed for PrP^c^ detection by means of immunohistochemistry (IHC) in the brain tissue of all the striped dolphins under study.

For *Brucella* spp. IHC, selected brain and spinal cord tissue sections from each animal were incubated with a MAb raised against the LPS antigen of *B. melitensis*, which had been validated by a previous study [4].

Adequate positive and negative control tissue samples were included in each WB and IHC run, with the latter ones being represented in IHC by brain and/or spinal cord tissue sections of *B. ceti*-infected striped dolphins from which the primary Ab had been either omitted or replaced by an unrelated Ab. 

The densitometric analysis of each sample was performed using the ImageJ 1.52 n program, which is an “open source” image processing program designed for multidimensional scientific images. We compared the signal of each single band obtained by means of WB analysis with the signal acquired through the glyceraldehyde-3-phosphate dehydrogenase (GAPDH) standard protein, with the obtained results being expressed as Relative Optical Densities (RODs). The WB data obtained were then submitted to statistical analysis with four repeated measurements, by means of the ANOVA test (Dunnet method), a method utilized for multiple comparisons with a control sample, which in our case was represented by the five *B. ceti*-uninfected dolphins.

## 3. Results

Microbiological investigations yielded *Brucella* spp. isolation from the brain tissue of 6 out of the 12 herein investigated cetaceans, with biomolecular (PCR) analyses giving positive results for *Brucella* spp. detection in the spleen and brain (*medulla oblongata*) from one additional striped dolphin individual. In this respect, however, it should be also noted that PCR does not represent the gold standard for the diagnosis of *Brucella* spp. infection. All the *Brucella* spp. isolates recovered were classified as *B. ceti* strains, with all the seven infected striped dolphins showing histomorphological evidence of neurobrucellosis-associated lesions in their brain and cervico-thoracic spinal cord segments. These lesions, generally quite severe, were characterized by various degrees/levels of magnitude in relation to the different brain areas, with a progressively growing intensity from the cranial to the caudal cerebral regions and to the neighboring spinal cord districts. More in detail, a subacute-to-chronic, non-suppurative, lympho-monocytic brain, and (cervico-thoracic) spinal cord meningitis, sometimes associated with a non-suppurative plexocoroiditis and focal gliosis, alongside occasional perivascular cuffing and/or spinal cord peri-endoganglioneuritis, were observed in all the herein investigated, *B. ceti*-infected, neurobrucellosis-affected striped dolphins.

As far as concerns the additional analyses performed on these dolphins, two *B. ceti*-infected individuals showed biomolecular evidence of *T. gondii* in their heart, with one of them also turning out to be bacteriologically positive for *Aeromonas sobria* in its lung and for *Clostridium glycolicum* in its gut. Furthermore, CeMV-specific genome sequences were identified in the CNS from another *B. ceti*-infected, neurobrucellosis-affected striped dolphin, with the successful recovery of *Aeromonas hydrophila* from the heart, liver, and kidney, as well as with simultaneous isolation of *Photobacterium damselae* from the lung and *Clostridium perfringens* from the intestine of two *B. ceti*-infected specimens.

By means of WB analysis, PrP^c^ IR was apparent in the brain tissue from all the herein investigated cetaceans. The PrP^c^ IR patterns evoked by the F89/160.1.5 MAb are represented in Figure 1 and Figure 2 (Appendix A Appendix A), both of which show the two peculiar PrP^c^ specific bands in a region between 35 kDa and 26 kDa. These were evident in all dolphin brain tissue samples analyzed, with three animals (two found stranded along the Italian coastline, one in the Canary Islands) also exhibiting a significantly increased PrP^c^ expression (2–3 times more) as compared to the five negative control samples (Figure 1 and Figure 2).

As expected, in the samples treated with proteinase K (PK) and deglycosidase F, the typical PrP^c^ bands disappeared (Figure 3 and Figure 4; Appendix A) thereby confirming the specific IR of F89/160.1.5 MAb only against PrP^c^. Despite the numerous and repeated attempts made, no positive IHC labeling for PrP^c^ could be obtained, by means of the aforementioned MAb, in any brain and spinal cord tissue slide from all the herein analyzed dolphins.

Finally, IHC investigations-which were carried out by means of the aforementioned anti-*B. melitensis* MAb produced at IZSAM-allowed us to detect *Brucella* spp.-associated/related antigens in the cytoplasm of numerous macrophagic/histiocytic cells composing the inflammatory infiltrates found in the CNS from the herein investigated, neurobrucellosis-affected dolphins (Figure 5), thus confirming the results of a previous study [4].

## 4. Discussion

The herein reported data, albeit preliminary and despite their intrinsic limitations, could serve as a reliable basis for future studies addressing the pathogenesis of neurobrucellosis in *B. ceti*-infected striped dolphins (as well as in other susceptible cetacean species), which is far from being elucidated. Despite their originality, the same data do not allow us to draw any firm conclusion about the role of PrP^c^ as a putative host cell receptor for *B. ceti*, which has been previously hypothesized elsewhere [17].

However, it should be duly emphasized that these data are the first ones ideally connecting, to the best of our knowledge, cerebral PrP^c^ expression with *B. ceti* infection in striped dolphins. Furthermore, despite its objective limitations, there is also an unequivocal “strength” in our study, which is provided by the good *post mortem* preservation degree (code 2, according to the popular classification scheme by Geraci and Lounsbury, 2005) [18] of all the 12 herein investigated striped dolphin specimens. Advanced *post mortem* autolysis is, in fact, a common finding when dealing with stranded cetacean specimens, with this undoubtedly representing a serious hurdle for several important laboratory exams like histopathological, IHC, WB, biomolecular and microbiological analyses.

Within this complex and intriguing framework, our study was inspired by the seminal work carried out several years ago on mice experimentally infected with *B. abortus*, in which a relevant pathogenetic role played by PrP^c^ in allowing bacterial entry into host’s macrophages was shown for the first time, with cell membrane PrP^c^ expression levels being inversely correlated to the phagocytic activity displayed by macrophages toward *B. abortus* [14]. Notwithstanding the above, coupled with an absolute lack of knowledge about the role of PrP^c^, if any, as a host cell receptor for *B. ceti* in striped dolphins, the interplay between the host’s immune cells and the corresponding *Brucella* spp. virulence factors, alongside the simultaneous PrP^c^ action(s)/effect(s), still remain largely unknown, if not even controversial. Indeed, while PrP^c^ mRNA silencing was shown to suppress cell antioxidant systems, it may also lead to an upregulation of pro-inflammatory cytokines like IL-12 and TNF-alfa, with *B. melitensis* infection apparently exerting no effects on bacterial phagocytosis [26]. While similar findings were reported in another study on macrophages infected with *B. suis* [27], the study by Watarai and coworkers [14] demonstrated that PrP^c^ would play a key role in the phagocytosis and bacterial transport of *B. abortus* into murine macrophages. Despite all these uncertainties and the caution deriving from the fact mouse is not the natural host of *B. abortus*, it is still believed that PrP^c^, in addition to its antioxidant properties, could play a role in *Brucella* spp. entry into host cells through phagocytosis inhibition/modulation [15].

Based upon a hypothetical model addressing *B. ceti* infection’s pathogenesis in neurobrucellosis-affected striped dolphins, an increased PrP^c^ biosynthesis could occur in the course of infection on the membrane of monocytes and macrophages, respectively carrying the bacterium in the bloodstream (monocytes) and allowing, thereafter, microbial neuroinvasion (macrophages). Following its entry into the cetacean host’s CNS, *B. ceti* could thus exert its peculiar neuropathogenic action, with subsequent development of neurobrucellosis-associated pathological changes. This hypothetical infection model could provide, among others, a plausible explanation for the different PrP^c^ expression profiles found in the brain tissue samples from the herein investigated, *B. ceti*-infected and neurobrucellosis-affected striped dolphins. Nevertheless, the variability in the cerebral PrP^c^ expression patterns among our striped dolphins could have a number of alternate/complementary explanations, with the intensity/magnitude of neurobrucellosis-associated lesions being just one of the possible options. Within this complex and intricate context, we should also consider, in fact, the role(s) played both by host-related factors (i.e., age, brain area/region sampled, physiological and/or simultaneously occurring pathological conditions, etc.) and by “extrinsic” factors (concurrent infections caused by other bacterial and/or non-bacterial microorganisms, alongside the effects of “persistent environmental pollutants” like dioxins, PCBs, “flame retardants”, heavy metals, micro/nanoplastics, etc.). Indeed, as “top predators” striped dolphins are known to accumulate and biomagnify heavy loads of environmental xenobiotics in their body tissues, with dioxins and dioxin-like compounds having been additionally reported to exert pro-apoptotic effects on human neuronal cells [28]. As a consequence, following dioxin-induced apoptosis, PrP^c^ overexpression could be plausibly expected to occur on CNS (brain and spinal cord) host cell membranes, given the anti-apoptotic role played by PrP^c^ [29].

As far as concerns the PrP^c^ specific IR seen in the brain tissue samples from the 12 striped dolphins under study, the three-dimensional structure of PrP^c^ consists of a globular C-terminal domain of over 100 amino acids (residues 121–231) and an N-terminal region of similar length (residues 23–120) but structurally flexible and disordered [10]. In many mammalian species, this globular structure is similar to human PrP^c^, as expected from the high percentage of sequence homology (data not shown) [30,31]. The polypeptide chain middle part (residues 106–126) is a highly conserved hydrophobic region and, in CNS cells, PrP^c^ is found in cholesterol-rich membrane microdomains (lipid rafts) [29,30,31,32]. Near this region, as already mentioned, there is the four amino-acid-bearing (142–145) PrP epitope (IHFG) which is specifically recognized by the F89.160.1.5 MAb. This region is poorly accessible to this anti-PrP MAb in sheep and goat samples, being at the same time highly reactive to the F99/97.6.1 MAb in the two aforementioned species. In this respect, a plausible explanation for the different IR patterns seen in our dolphin brains following F89.160.1.5 MAb utilization, as compared to caprine and ovine CNS samples, could be the impact(s) exerted by a more or less “superficial” location of a single amino acid between these three species at the level of the PrP IHFG epitope. Additionally, while the three-dimensional β-sheet structure of the pathological PrP isoform, PrP^Sc^, gives it less accessibility to the PK hydrolysis site, PrP^c^ consists of a folded terminal COOH domain with three α-helices and a free terminal NH_2_ segment in solution, thus being more easily accessible by proteases [29]. The role of PrP^c^ as a precursor of PrP^Sc^ during the onset of PDs/TSEs has been extensively investigated, with PrP^c^ additionally playing an important role in T lymphocytes’ and phagocytes’ functioning, thereby acting as an immune response regulator under both physiological and pathological conditions. Furthermore, PrP^c^ has been also identified as a potential pathogenetic factor and as a biomarker for other neurodegenerative disorders like Alzheimer’s disease, as well as for bacterial and viral infections [8,9]. Nakato et al. [13] report that, following oral infection, *B. abortus* is effectively internalized only in microfold cells (M cells), attributing to them the role of “gateway” for this bacterium, which uses PrP^c^ located on their apical surface as an uptake receptor. The same investigators have additionally reported that PrP^c^ plays an important role in the uptake of *B. abortus* by macrophages [13]. Another study showed an upregulation of PrP^c^ mRNA in murine bone marrow cell cultures challenged with *Escherichia coli* in the early stages of bacterial infection [33]. The PrP^c^-encoding gene (*PRNP*) could also interact with the transcription factors of heat shock proteins (HSPs). For example, when *Brucella* spp. invades phagocytes, HSPs have been reported to promote the aggregation of PrP^c^ on their surface; moreover, PrP^c^ has been reported to colocalize with the growth factor protein Grb2, which initiates the rearrangement of the cytoskeleton and regulates the control of signaling pathways [8,9,10,11,12,13,14,15,16,17,18,19,20,21,22,23,24,25,26,27,28,29,30,31,32,33,34]. Microglial cells are phagocytes clustered in the brain, which can be activated during brain trauma, infections, and neurodegenerative diseases. Although they are known to become activated in the early stages of Creutzfeldt-Jakob disease, the functional relationships between microglia and PrP^c^ remain unclear; however, the activation of these cells is accompanied by a downregulation of the expression of the *PRNP* gene and it also appears that, during infection or in degenerative CNS disorders, this could promote the conversion of PrP^c^ into PrP^Sc^ [8].

From the densitometric analysis performed on the bands obtained with the F89.160.1.5 MAb, a strong and statistically significant increase in the intensity of the two bands, namely 35 kDa and 26 kDa, was found in the brain tissue from three out of the seven *B. ceti*-infected, neurobrucellosis-affected striped dolphins under study. While these results do not allow us to draw any firm conclusion(s) about a putative role of PrP^c^ as host cell receptor for *B. ceti* [17], it seems of interest that the inflammatory infiltrates seen in the brain and in the cervico-thoracic spinal cord segments from our *B. ceti*-infected, neurobrucellosis-affected striped dolphins were densely populated by macrophagic/histiocytic cells often harboring *Brucella* spp. antigen in their cytoplasm, similarly to what was reported in macrophages from mice experimentally infected with *B. abortus* [14].

Notwithstanding the above, however, the existence of pathological and pathogenetic correlations, if any, between the magnitude and the extent of *B. ceti*-associated lesions, on one side, and the expression levels of PrP^c^ within different CNS areas, on the other, should be thoroughly assessed in future studies.

## 5. Conclusions

The herein reported data, albeit preliminary, despite their intrinsic limitations, could serve as a reliable basis for future studies addressing the pathogenesis of neurobrucellosis in *B. ceti*-infected striped dolphins (as well as in other susceptible cetacean species), which is far from having been elucidated. Nevertheless, despite their originality, the same data do not allow us to draw any firm conclusion about the role of PrP^c^ as a putative host cell receptor for *B. ceti*. In this respect, an upregulation of PrP^c^ mRNA in the brain parenchyma of neurobrucellosis-affected striped dolphins could be hypothesized during the different stages of *B. ceti* infection, in a similar fashion to what already shown in murine bone marrow cells challenged with *Escherichia coli* [33].

Noteworthy, the inflammatory infiltrates seen in the brain and in the cervico-thoracic spinal cord from the herein investigated, *B. ceti*-infected, neurobrucellosis-affected striped dolphins were densely populated by macrophage/histiocyte cells often harboring *Brucella* spp. antigen in their cytoplasm, similarly to what was reported in macrophages from mice experimentally challenged with *B. abortus* [14]. This could add support to the hypothesis of a PrP^c^ role in the neuropathogenesis of *B. ceti* infection in neurobrucellosis-affected striped dolphins [17], as previously shown in murine macrophages following experimental *B. abortus* infection [14]. Hypothetically, an increased PrP^c^ biosynthesis could be expected to occur during *B. ceti* infection on the membrane of monocytes and macrophages, carrying the microbial pathogen in the bloodstream (monocytes) and allowing, thereafter, bacterial neuroinvasion (macrophages). Following its entry into the cetacean host’s CNS, *B. ceti* could then exert its peculiar neuropathogenic action, with subsequent development of neurobrucellosis-associated pathological changes. This could provide, among others, a plausible explanation for the different PrP^c^ expression profiles found in the brain tissue samples from the herein investigated, *B. ceti*-infected and neurobrucellosis-affected striped dolphins. Notwithstanding the above, the variability in the cerebral PrP^c^ expression patterns among our striped dolphins could have a number of alternate/complementary explanations, with the intensity/magnitude of neurobrucellosis-associated lesions being just one out of the plausible/possible options. Within this complex and intricate context, we should also consider, in fact, the role(s) played both by host-related factors (i.e., age, brain area/region sampled, physiological and/or simultaneously occurring pathological conditions, etc.) and by “extrinsic” factors (concurrent infections caused by other bacterial and/or non-bacterial microorganisms, alongside the effects of “persistent environmental pollutants” like dioxins, PCBs, “flame retardants”, heavy metals, micro/nanoplastics, etc.). Indeed, as “top predators” striped dolphins are known to accumulate and biomagnify heavy loads of environmental xenobiotics in their body tissues, with dioxins and dioxin-like compounds having been additionally reported to exert pro-apoptotic effects on human neuronal cells [28]. As a consequence, following dioxin-induced apoptosis, PrP^c^ overexpression could be plausibly expected to occur on CNS (brain and spinal cord) host cell membranes, given the anti-apoptotic role played by PrP^c^ [29].

## Figures and Tables

**Figure 1 animals-12-01304-f001:**
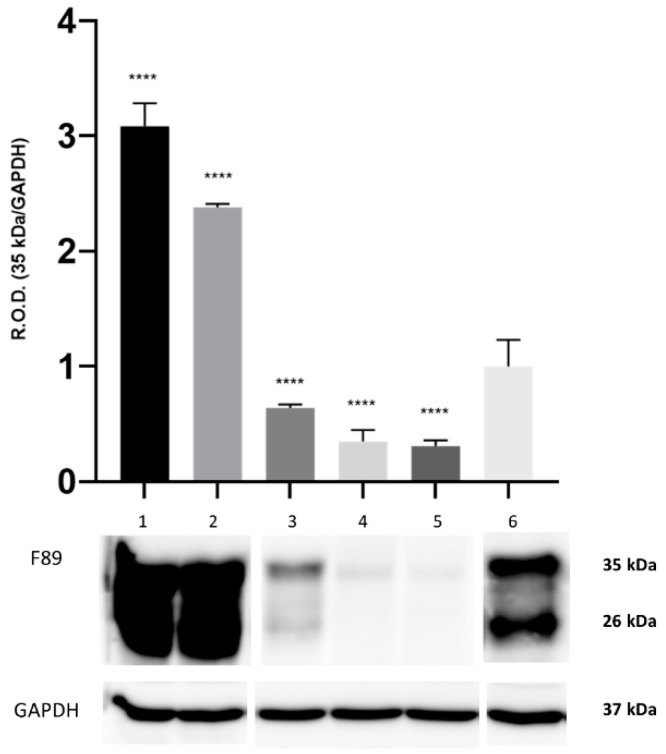
Representative Western blot (WB) analysis of the 35 kDa and 26 kDa bands obtained with the anti-prion protein (PrP) MAbF89/160.1.5. Brain tissue from three striped dolphin specimens found stranded along the Italian coast of Apulia (lanes 1 and 3) and in Canary Islands (lane 2), all of which affected by neurobrucellosis due to *Brucella ceti*. The results of WB analyses carried out on the brain tissue of one lamb (lane 4), one goat (lane 5), and one control (*B. ceti*-negative) striped dolphin found stranded along the Tyrrhenian coast of Tuscany (lane 6) are also shown (original Western blot figure is in Appendix A). GAPDH was used as a protein load control. The graph depicts only the optical density of the 35 kDa band. The results are expressed as mean ± SEM (*n* = 4). The statistical significance was analyzed by the ANOVA Dunnett method, **** *p* < 0.0001 vs. the striped dolphin negative control specimens.

**Figure 2 animals-12-01304-f002:**
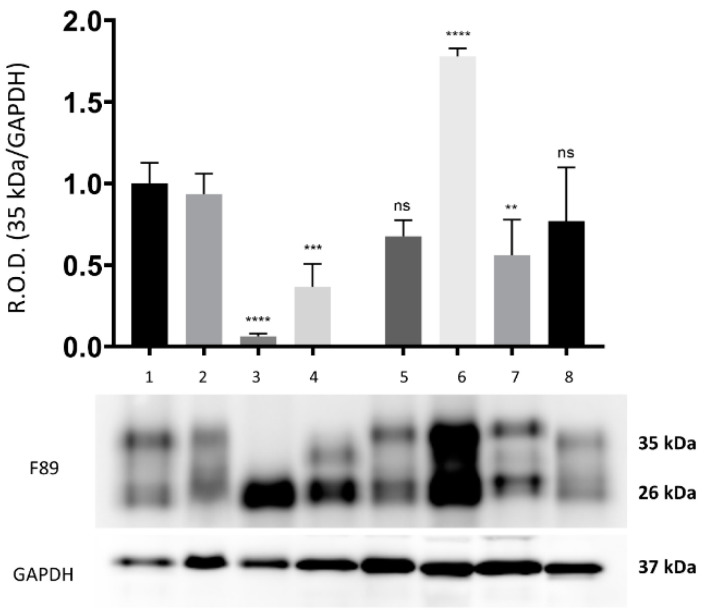
Representative WB analysis of the 35 kDa and 26 kDa bands obtained with the anti-PrP MAbF89/160.1.5. Brain tissue samples from 8 striped dolphins found stranded along the Italian coastline (lanes 1–8), four of which affected by neurobrucellosis due to *Brucella ceti* (lanes 5–8). The results of WB analyses carried out on the brain tissue of four control (*B. ceti*-negative) striped dolphins found stranded along the Italian coastline (lanes 1–4) are also shown (original Western blot figure is in Appendix A). GAPDH was used as a protein load control. The graph depicts only the optical density of the 35 kDa band. The results are expressed as mean ± SEM (*n* = 4). The statistical significance was analyzed by ANOVA Dunnett method, **** *p* ≤ 0.0001, *** *p* ≤ 0.001, ** *p* ≤ 0.01, ns *p* ≥ 0.05 vs. the striped dolphin negative control specimens.

**Figure 3 animals-12-01304-f003:**
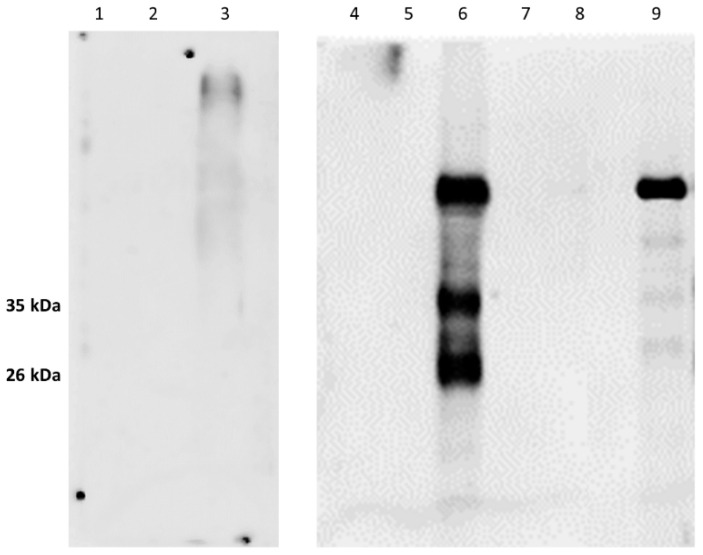
Results of WB analyses carried out with the anti-PrP MAb F89/160.1.5 on the pellets obtained after ultracentrifugation of brain tissue samples from one goat (lanes 1–3), as well as from a *B. ceti*-infected striped dolphin stranded along the Italian coast of Apulia (lanes 4–6) and from one lamb (lanes 7–9). The typical 35 kDa and 26 kDa PrP^c^ bands are shown at lanes 3, 6 and 9 after ultracentrifugation, while lanes 2, 5 and 8 show the WB results obtained after proteinase K (PK) digestion, and lanes 1, 4 and 7 after treatment with deglycosydase F (original Western blot figure is in Appendix A).

**Figure 4 animals-12-01304-f004:**
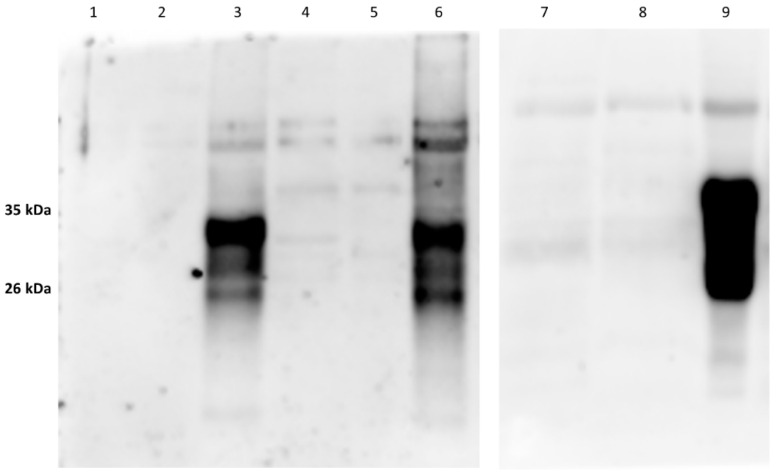
Results of WB analyses carried out with the anti-PrP MAb F89/160.1.5 on the pellets obtained after ultracentrifugation of brain tissue samples from one negative control (*B. ceti*-uninfected) striped dolphin found stranded along the Tyrrhenian coast of Tuscany (lanes 1–3), as well as from two *B. ceti*-infected, neurobrucellosis-affected striped dolphin specimens found stranded along the Italian coast of Apulia (lanes 4–6) and in Canary Islands (lanes 7–9), respectively. The typical 35 kDa and 26 kDa PrP^c^ bands are shown at lanes 3, 6 and 9 after ultracentrifugation, while lanes 2, 5 and 8 show the WB results obtained after PK digestion, and lanes 1, 4 and 7 after treatment with deglycosydase F (original Western blot figures are in Appendix A).

**Figure 5 animals-12-01304-f005:**
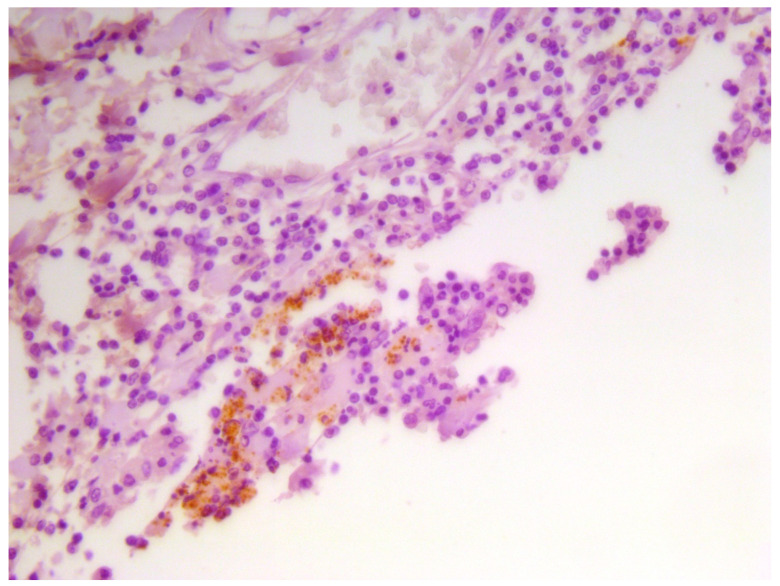
*B. ceti*-infected, neurobrucellosis-affected striped dolphin (*S. coeruleoalba*) specimen. Cervical spinal cord. Evidence of *Brucella* spp. antigen in the cytoplasm of macrophages/histiocytes forming the inflammatory infiltrate is shown in this picture. Immunohistochemistry (IHC) against *Brucella* spp. with an anti-*B. melitensis* MAb (produced at IZSAM “G. Caporale”, Teramo, Italy), Mayer’s hematoxylin counterstain.

## Data Availability

All the data reported herein are original and are being made publicly available for the first time through the present manuscript.

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
