# Peer review of "Cellular Prion Protein Expression in the Brain Tissue from Brucella ceti-Infected Striped Dolphins (Stenella coeruleoalba)"

_animals, 2022, doi:10.3390/ani12101304_

Round 1

Reviewer 1 Report

The authors present results for initial investigations of PrPc expression in the brains of stranded striped dolphins, utilizing samples obtained at necropsy.  Below are some comments and suggestions that could improve the manuscript. Overall, this information is of value though could benefit from expansion in some areas.

The data is limited to western blot analysis given the complications of PrPc IHC analysis that described in the abstract, though not in the methods or results (?). Consider including description of method for PrPc IHC and the results (even if negative or inconclusive).

The presence of Brucella sp. was determined in the lesions via IHC for some cases - were any of these cases confirmed with PCR of brain (or other tissues)? Were histological lesions consistent with neurobrucellosis in all of the infected cases and were these lesions similarly severe? Some description of the cases, since there are few, and perhaps some sort of lesion scoring and/or explanation of case criteria would be beneficial. 

Instead of describing cells as "macrophage-like" cells, consider calling them macrophages or if unclear of histogenesis, use a macrophage marker (Iba-1, CD11, CD68, etc). 

Besides IHC, there are other methods to demonstrate RNA or protein expression in tissues (in situ hybridization). Was this considered?

Author Response

Comments and Suggestions for Authors

The authors present results for initial investigations of PrPc expression in the brains of stranded striped dolphins, utilizing samples obtained at necropsy.  Below are some comments and suggestions that could improve the manuscript. Overall, this information is of value though could benefit from expansion in some areas.

The data is limited to western blot analysis given the complications of PrPIHC analysis that described in the abstract, though not in the methods or results (?). Consider including description of method for PrPIHC and the results (even if negative or inconclusive).

Reply: We really appreciate these valuable and precious comments, remarks and suggestions kindly made by this Reviewer. In this respect, we have expanded both the “Materials and Methods” and the “Results” sections of our manuscript (ms), thereby including into them the immunohistochemistry (IHC) methodology used and the IHC results obtained.

The presence of Brucella sp. was determined in the lesions via IHC for some cases - were any of these cases confirmed with PCR of brain (or other tissues)? Were histological lesions consistent with neurobrucellosis in all of the infected cases and were these lesions similarly severe? Some description of the cases, since there are few, and perhaps some sort of lesion scoring and/or explanation of case criteria would be beneficial. 

Reply: We really appreciate these valuable and precious comments, remarks and suggestions kindly made by this Reviewer. In this respect, biomolecular (PCR) confirmation of Brucella spp. infection was obtained from both the spleen and the brain (medulla oblongata) tissues of one striped dolphin specimen included in our study. Furthermore, the microscopic lesions detected in various central nervous system (CNS) regions and districts (brain and spinal cord) from the herein investigated striped dolphins were consistent with those commonly seen/reported in Brucella ceti-infected, neurobrucellosis-affected striped dolphins (subacute-to-chronic, non-suppurative, lympho-monocytic brain and spinal cord meningitis; non-suppurative plexocoroiditis; focal gliosis; perivascular cuffing; spinal cord peri-endoganglioneuritis). A quite prominent CNS lesions’ severity (lesion scoring) was generally observed in the striped dolphins under study, with all this information having been adequately included into the present ms revision, as wisely suggested by the Reviewer.

Instead of describing cells as "macrophage-like" cells, consider calling them macrophages or if unclear of histogenesis, use a macrophage marker (Iba-1, CD11, CD68, etc). 

Besides IHC, there are other methods to demonstrate RNA or protein expression in tissues (in situ hybridization). Was this considered?

Reply: With reference to the cells composing the inflammatory infiltrates seen in the brain and spinal cord from the herein investigated, neurobrucellosis-affected striped dolphins, the expression “macrophage-like cells” has been replaced by “macrophages/histiocytes” in the concerned Figure’s legend. In this respect, while we greatly appreciate the valuable suggestions made by the Reviewer, it should be duly emphasized that the aforementioned cells’ morphology in haematoxylin-eosin (HE)-stained sections is unequivocally consistent with that of macrophages/histiocytes. Additionally, although time- and finance-related constraints do not allow us to perform ad hoc IHC investigations against the biomarkers usefully suggested by the Reviewer (Iba-1, CD11, CD68, etc.), it should be additionally underlined that previous IHC studies carried out by means of an anti-MHC Class II human monoclonal antibody (MAb) have already allowed us to characterize as macrophages/histiocytes the inflammatory cells found in the striped dolphin specimen stranded in 2004 in Canary Islands (Proietto et al., 2008). The results of this study were in tight agreement with those reported in a previous work (Jaber et al., 2003). Also this information has been duly incorporated into the current ms revision. Finally, as far as concerns the cellular prion protein (PrPc) expression in the brain tissue from the herein investigated striped dolphins, our study was focused on the protein’s expression, which was evaluated by means of both Western-blot (WB) analysis (successful) and IHC (unsuccessful). To this aim, while the simultaneous evaluation of PrPc expression through the analysis of “messanger RNA (mRNA)” could have provided valuable additional data from a “dynamic/kinetic standpoint”, we do not believe that the results of “in situ hybridisation”-based/related investigations would have allowed us to gain further information within such context, given their objective limitations for an accurate assessment of protein expression levels.  

Reviewer 2 Report

The manuscript by Angelucci et al. includes a study investigating the cellular prion protein (PrPc) expression in the brain tissue from Brucella ceti-infected striped dolphins.  In general, the article is original, however, there is no evidence of in vitro conditions where all factors are controlled, versus the individuals used for this study that are free-living dolphins and many conditions are present related to stranding and dead. From the title, it should be evident that what is detected is the presence of PrpC in the nervous tissue of striped dolphins, but the same information generated in animals does not conclude and contradicts the expression of the protein in animals without the infection. I believe that the authors should rethink the scope of their data and redefine their conclusions in this regard since they do not rule out the sole expression by natural activities of the protein under physiological conditions. Therefore, caution should be made with subjective conclusions without proven data

Please consider the following points:

  1. Since the role of prion proteins in neuronal Brucella infection, specifically in the interaction of the pathogen and the host cell is controversial, the conclusions of this article must be interpreted with caution and re-worded.

  1. Previous studies have proved that silencing of PrPC mRNA by small interfering RNA does not have a significant effect on prion proteins on phagocytosis and intracellular killing of B. melitensis in microglia cells (Erdogan, 2013)

  1. The methodology of this article only confirm the presence of the PrPC in the neurological tissue of striped dolphin, however, there is no methodology described here that suggest a role in the neuropathogenesis of Brucella ceti and therefore the conclusions need to be changed.

  1. Other bacterial pathogens potentially present in the brain of these animals are not discarded nor mentioned any methodology used to detect them.

  1. Since cellular prion proteins (PrPC) have some crucial biological functions including roles in neurotransmission, signal transduction, programmed cell death, antioxidant effects, and a role in the endocytosis of bacteria within cells, how do the authors correlate the expression of PrPC with the infection of only Brucella ceti and not other biological function of the same infection with this pathogen or other pathogen potentially present on these brains?. This protein has different biological functions and its single expression in both infected and not infected animals only represents its presence. All the conclusions of the expression because of Brucella infection of this study must be deeply analyzed and reword 

  1. Other causes of the high expression of PrPC in animals must be included. Brains of striped dolphins with other bacterias such as Listeria monocytogenes have been described in Italy and this kind of material need to be included since this overexpression in the infected brain with other pathogens can probably occur.

  1. The single-use of Proteinase K (PK) and deglycosidase F, to confirm the identity of the protein is not enough for confirming the specific reactivity of F89/160.1.5 MAb only against PrPc. This antibody needs to be tested and prove no cross-reactivity with other similar proteins. Especially with the results of IHQ in this study. Proteomics can confirm this Identity If the authors or the manufacturers have more information of this specificity, this information needs to be included in the discussion.

  1. There is no information on the specificity of the anti- melitensisMab in the IHC brains infected with other gram-negative bacteria. If the authors or the producers have more information of this specificity, this information needs to be included in the discussion.

  1. Line 96-97. The methodology to confirm ceti infection must be included. If these animals are part of other articles, the references must be also included. The methodology to confirm the dolphins were negative for B. ceti must be included.
  2. Neurobrucellosis is not always present in  ceti infected striped dolphins.

  1. Line 96-101. What was the decomposition grade on these tissues? Include the codes of decomposition for each animal. The stranded animals arrived alive? All this information is important for the quality of the samples and the normal proteolysis within the brain that have a direct impact on the final analysis of expression or detection level in WB. 

  1. Line 138-146. The different intensity of the color gray means something? The different colors can be useful to differentiate between infected and not infected. 

  1. Line 138-146. This figure confirms the higher expression of PrPc is higher in non-infected animals to some of the infected animals. This graph needs to be deeply discussed accordantly in the conclusions

  1. Line 156. Ultracentrifugation?

  1. Line 157 Figure 3 

  1. The typical 35 kDa and 26 kDa PrPc bands are not shown at lane 3

  1. Line 169 Figure 4. The figure must include the negative control of the same slide by IHQ.  

  1. Line 170. This IHQ has been tested for other gram-negative bacteria to rule out the cross-reactivity with other pathogens

  1. Line 204-205 This sentence needs to be reworded as discussed above, there is not enough information within this study to elucidate this subjective conclusion. Other bacteria or conditions are not studied and other factors can cause this expression. No direct evidence of the role of B ceti in this expression in brain tissue is included (see comment No.2)

  1. Line 216-220. Neurobrucellosis is a chronic infection of bacterias of Brucella genera in the different infected species, no necessarily correlated with the level of inflammation or tissue damage, therefore the comparison between in vitro conditions and in vivo conditions of other gram-negative bacteria needs to be fully discussed with a number o possible variants affecting the expression of the infected animals versus higher expression in not infected animals.

  1. Line 244. This sentence needs to be reworded as discussed above, there is not enough information within this study to elucidate this subjective conclusion. Other bacteria or conditions are not studied and other factors can cause this expression. No direct evidence of the role of B ceti in this expression in brain tissue is included (see comment No.2) the comparison between in vitro conditions and in vivo conditions of other gram-negative bacteria needs to be fully discussed with a number o possible variants affecting the expression of the infected animals versus higher expression in not infected animals.

  1. Line 248-250. This conclusion is contrary to the information shown in Figure 2

  1. Line 250-260 See comment 2 This conclusion is contrary to the information shown in Figure 2

Author Response

Comments and Suggestions for Authors

The manuscript by Angelucci et al. includes a study investigating the cellular prion protein (PrPc) expression in the brain tissue from Brucella ceti-infected striped dolphins.  In general, the article is original, however, there is no evidence of in vitro conditions where all factors are controlled, versus the individuals used for this study that are free-living dolphins and many conditions are present related to stranding and dead. From the title, it should be evident that what is detected is the presence of PrpC in the nervous tissue of striped dolphins, but the same information generated in animals does not conclude and contradicts the expression of the protein in animals without the infection. I believe that the authors should rethink the scope of their data and redefine their conclusions in this regard since they do not rule out the sole expression by natural activities of the protein under physiological conditions. Therefore, caution should be made with subjective conclusions without proven data.

Reply: We really appreciate the highly valuable and precious comments, remarks and suggestions kindly provided by this Reviewer. In this respect, while the Reviewer correctly states that “there is no evidence of in vitro conditions where all factors are controlled, versus the individuals used for this study that are free-living dolphins and many conditions are present related to stranding and death”, it should be duly emphasized, on the other side, that when dealing with stranded cetaceans, we are not coping (by definition!) with “normal animals”, but rather with animals whose health condition has been compromised by the intervention, throughout a more or less prolonged amount of time, of a number of disease-causing agents  - many of which of “anthropogenic origin” - of either physical, chemical or biological nature. Furthermore, cetaceans - with special emphasis on Odontocetes, as in the case of the herein investigated striped dolphins - are “top predators”, thereby giving rise to “bioaccumulation and biomagnification” processes of a huge number and variety of chemical environmental pollutants in their body tissues, which may result in their reproductive function/biology and immune response impairment. Should “anybody” ask if there is an “ideal biological control” among wild, free-living cetaceans, the answer would probably be no (with partial exceptions, though only partial, being likely represented by “healthy” cetaceans caught in fishing nets). By contrast, when dealing with “conventional” animals, for which many “normal” values are available, a condition of “biological health/normality” appears to be much easier to get defined. Notwithstanding the above, and even more important, the strategic/crucial role of stranded cetaceans as “sentinels/biomonitors” for the health and conservation status of their “conspecifics and heterospecifics” living in the open sea should be adequately underscored, based upon the objective fact that stranded cetacean specimens provide the Scientific Community with an unique chance to study the health and conservation of wild cetaceans inhabiting planetary oceans and seas, whose health and conservation status appear to be increasingly threatened, with a number of cetacean species and populations worldwide now facing a progressively growing extinction risk.

This is the framework/context inside which any study focused on cetaceans’ pathology and, more in general, on cetaceans’ health and conservation assessment, should be unavoidably (and correctly) placed.

As already mentioned, we are deeply indebted to this Reviewer, whose comments, concerns, remarks and suggestions have greatly helped us improve the overall quality of our ms, which has now been carefully revised thereby taking into special account all of the aforementioned comments, concerns, remarks and suggestions.

More in detail, being PrPc a constitutively expressed protein, with its expression levels being particularly consistent in the CNS from all animal species and whose function(s) having been only partially elucidated thus far, we have “rethought the scope(s) of our data”, thereby “redefining (our ms) conclusions”, since PrPc expression is known to be influenced/modulated by “physiological conditions” and, presumably, also by a number of simultaneously occurring pathological conditions (either known or unknown).

Please consider the following points:

  1. Since the role of prion proteins in neuronal Brucella infection, specifically in the interaction of the pathogen and the host cell is controversial, the conclusions of this article must be interpreted with caution and re-worded.

Reply: As correctly suggested by the Reviewer, given the controversial role of PrPc in Brucella spp. CNS infection, we have carefully re-written the ms conclusions, thereby citing/quoting additional references, as usefully adviced also by Reviewer 3.   

  1. Previous studies have proved that silencing of PrPC mRNA by small interfering RNA does not have a significant effect on prion proteins on phagocytosis and intracellular killing of B. melitensis in microglia cells (Erdogan, 2013).

Reply: We would like to thank both this Reviewer and Reviewer 3 for kindly suggesting us to cite/quote the interesting paper by Dr Erdogan (2013). In this respect, Brucella spp. entry into macrophages/histiocytes and the host’s response and other defense mechanisms against this microbial pathogen have not been clearly elucidated, thus far. More in detail, while PrPc mRNA silencing suppressed cell antioxidant systems, it also led to an upregulation of pro-inflammatory cytokines like IL-12 and TNF-alfa, with B. melitensis infection apparently exerting no effects on bacterial phagocytosis (Erdogan, 2013). Similar findings were reported in another study on macrophages infected with B. suis (Fontes et al., 2005), in contrast to the results of Watarai et al. (2003), who demonstrated that PrPc has relevant effects on phagocytosis and bacterial transport in B. abortus-infected macrophages.

In neurobrucellosis, however, the interaction mechanisms between the host cell and the corresponding Brucella spp. virulence factors that force the oxidative defense system of the host cells to react, together with the simultaneous PrPc action(s)/effect(s), remain largely unknown. Despite these unknowns, it is believed that PrPc, in addition to its antioxidant properties, could still play a role in Brucella spp. entry into cells through phagocytosis modulation/inhibition (Aydin et al., 2013), although we do not know yet if and to what extent PrPc mediates B. ceti entry into cetacean (striped dolphin) host’s macrophages/histiocytes.

All these thoughts and concepts have been duly included into the present ms revision, the conclusions of which have been carefully re-written, as wisely suggested by this Reviewer.

Erdogan, S., Duzguner, V., Kucukgul, A., Aslantas, O. Silencing of PrPC (prion protein) expression does not affect Brucella melitensis infection in human derived microglia cells. Research in Veterinary Science 95 (2013) 368-373.

Fontes, P.; Alvarez-Martinez, M.T.; Gross, A.; Carnaud, C.; Kohler, S.; Liautard, J.P. Absence of Evidence for the Participation of the Macrophage Cellular Prion Protein in Infection with Brucella suis. Infection and Immunity, 2005, Vol. 73, No. 10, pages 6229–6236. doi:10.1128/IAI.73.10.6229-6236.2005.

Aydin, M.; Gilmore, D.F.; Erdogan, S.; Duzguner, V.; Ahn S. The Role of Cellular Prion Proteins (PrPC) on Microglial Brucella Infections. Agric. Food Anal. Bacteriol. 2013, Vol 3, pages 268-280.

  1. The methodology of this article only confirm the presence of the PrPC in the neurological tissue of striped dolphin, however, there is no methodology described here that suggest a role in the neuropathogenesis of Brucella ceti and therefore the conclusions need to be changed.

Reply: The ms conclusions have been duly changed, as usefully adviced by the Reviewer.

  1. Other bacterial pathogens potentially present in the brain of these animals are not discarded nor mentioned any methodology used to detect them.

Reply: We have now incorporated all this relevant information (including the concerned laboratory methodology used) into the current ms revision, as wisely suggested by the Reviewer.

  1. Since cellular prion proteins (PrPC) have some crucial biological functions including roles in neurotransmission, signal transduction, programmed cell death, antioxidant effects, and a role in the endocytosis of bacteria within cells, how do the authors correlate the expression of PrPC with the infection of only Brucella ceti and not other biological function of the same infection with this pathogen or other pathogen potentially present on these brains?. This protein has different biological functions and its single expression in both infected and not infected animals only represents its presence. All the conclusions of the expression because of Brucella infection of this study must be deeply analyzed and reworded.

Reply: Again, we would like to express our most sincere feelings of appreciation and gratitude to this Reviewer for her/his highly valuable and precious concerns, comments, remarks and suggestions, which have been carefully and adequately taken into account in the present ms revision, with special emphasis on its conclusions (see also reply to point 2).

  1. Other causes of the high expression of PrPC in animals must be included. Brains of striped dolphins with other bacterias such as Listeria monocytogenes have been described in Italy and this kind of material need to be included since this overexpression in the infected brain with other pathogens can probably occur.

Reply: The Reviewer is absolutely correct when making reference to a previously published study describing “neurolisteriosis” (by Listeria monocytogenes) in association with a Brucella spp. and Toxoplasma gondii coinfection in a striped dolphin specimen found stranded on the Italian coastline in 2015. Nevertheless, this animal was not included among the herein investigated striped dolphins, on which both “direct” and “indirect” laboratory investigations against viral (Cetacean Morbillivirus), protozoan (Toxoplasma gondii) and bacterial (both aerobic and anaerobic) pathogens were also carried out, as duly outlined/specified in the current ms revision.

  1. The single-use of Proteinase K (PK) and deglycosidase F, to confirm the identity of the protein is not enough for confirming the specific reactivity of F89/160.1.5 MAb only against PrPc. This antibody needs to be tested and prove no cross-reactivity with other similar proteins. Especially with the results of IHQ in this study. Proteomics can confirm this Identity If the authors or the manufacturers have more information of this specificity, this information needs to be included in the discussion.

Reply: With reference to this relevant point, we would like to inform you that, in order to confirm the specific identity of the “cellular prion protein” (PrPc) in the brain tissue from the herein investigated striped dolphins, we considered it appropriate, before starting our work, to analyze the PrPc primary aminoacidic sequence in striped dolphins as compared to the human, caprine and ovine PrPc primary sequences (see image below). This was done with the chief objective to comparatively evaluate, in the four mammalian species cited above,  the localization of the epitopes specifically recognized by the F89.160.1.5 Mab (see image below).

Indeed, in many mammalian species the PrPc globular structure is identical to that of the human prion protein, as expected from the high percentage of primary sequence homology (Lopez et al., 2005; Lysek et al., 2005). The central part of the polypeptide chain is a highly conserved hydrophobic region. Furthermore, in this region there is the epitope (IHFG) specifically recognized by the F89.160.1.5 MAb, which also recognizes both the “normal” and the “pathological” PrP isoforms in tissues from sheep, cattle, mule deer, elk, white-tailed deer and humans.

Lopez-Garcia F, Zahn R, Riek R, Wuthrich K. NMR structure of the bovine prion protein. Proc. Natl. Acad. Sci. USA. 2005; 97: 8334-8339.

Lysek DA, Schorn C, Nivon LG, Esteve-Moya V, Christen B, Calzolai L, von Schroetter C, Fiorito F, Herrmann T, Guntert P, Wuthrich K. Prion protein NMR structures of cats, dogs, pigs, and sheep. Proc. Natl. Acad. Sci. USA. 2005; 102: 640-645.

All the above information has now been included into the current ms revision.

  1. There is no information on the specificity of the anti-B. melitensis Mab in the IHC brains infected with other gram-negative bacteria. If the authors or the producers have more information of this specificity, this information needs to be included in the discussion.

Reply: This information has now been included in the “Discussion” section of the present ms revision. However, no apparent cross-reactivities with other Gram-negative bacteria were found when performing IHC analyses with the anti-B. melitensis MAb utilized in this work, as previously reported in an ad hoc validation study published elsewhere (Di Francesco et al., Vet. Ital., 2019).

  1. Line 96-97. The methodology to confirm B. ceti infection must be included. If these animals are part of other articles, the references must be also included. The methodology to confirm the dolphins were negative for B. ceti must be included.

Reply: The concerned methodology has been duly included into the current ms revision, as usefully suggested by the Reviewer.

  1. Neurobrucellosis is not always present in B. ceti infected striped dolphins.

Reply: This is absolutely true. Nevertheless, all of the Brucella ceti-infected striped dolphin specimens under study were also affected by a more or less severe (generally quite severe) and simultaneously occurring “neurobrucellosis disease condition”. This information has been duly incorporated into the present ms revision.

  1. Line 96-101. What was the decomposition grade on these tissues? Include the codes of decomposition for each animal. The stranded animals arrived alive? All this information is important for the quality of the samples and the normal proteolysis within the brain that have a direct impact on the final analysis of expression or detection level in WB. 

 Reply: The Reviewer is absolutely correct, being the “preservation/conservation code” of each cetacean specimen included in any post mortem investigation (with special reference to “anatomo-histopathological, immunohistochemical, immunobiochemical and microbiological analyses”, as in the herein dealt striped dolphins) a “prerequisite” of key relevance. In this respect, all of the herein investigated striped dolphin specimens were found stranded lifeless, in a good/fresh “preservation/conservation status” (code 2, according to the classification scheme by Geraci & Lounsbury, 2005). All this information has been duly included into the current ms revision.

  1. Line 138-146. The different intensity of the color gray means something? The different colors can be useful to differentiate between infected and not infected. 

Reply: The “different intensity of the color gray” in WB pictures does not have any specific meaning. If the Reviewer agrees, we would leave the aforementioned images’ color as it currently is.

  1. Line 138-146. This figure confirms the higher expression of PrPc is higher in non-infected animals to some of the infected animals. This graph needs to be deeply discussed accordantly in the conclusions

Reply: As wisely suggested by the Reviewer, we have carefully reconsidered these issues in the revised ms “conclusions”.

  1. Line 156. Ultracentrifugation?

Reply: As far as this point is specifically concerned, we would like to emphasize that “ultracentrifugation” is a step to be necessarily performed for an adequate brain tissue samples’ preparation before their investigation/study by means of WB analysis (Nonno et al., 2003, J. Clin. Microbiol.).

  1. Line 157 Figure 3 

Reply: See reply below, please.

  1. The typical 35 kDa and 26 kDa PrPc bands are not shown at lane 3

Reply: The PrPc bands to which the Reviewer makes reference in Figure 3 are from the brain tissue of a goat, which was used as a control sample. Indeed, as also specified in the original version/text of our ms, “this PrPc region appears to be poorly accessible (in sheep and goat) to the anti-PrP F89.160.1.5 Mab (utilized in our study)”. As still reported in our ms, “a plausible explanation for the different IR patterns seen in our dolphin brain tissue samples, following F89.160.1.5 MAb utilization, could lie upon the impact(s) exerted by a more or less “superficial” location of the aminoacidic differences occurring, at the level of the MAb-specific epitope, between the striped dolphin’s sequence and those of sheep and goat”.

Anyway, we leave to the Reviewer’s decision/discretion the possibility to maintain “lane 3” in Figure 3 as it currently stands or, alternatively, to delete it.

  1. Line 169 Figure 4. The figure must include the negative control of the same slide by IHQ.  

Reply: We would like to thank the Reviewer for her/his suggestion. In this respect, my co-Authors and I do not understand, honestly speaking, if this Reviewer makes reference to Figure 4 (dealing with WB analysis), or to Figure 5 (dealing with IHC). Should the latter be the case, let me comment, as a “Board-Certified Veterinary Pathologist”, that it is very unusual to see micrographs of “negative control tissue specimens” in any published manuscript. As a matter of fact, appropriate “positive” and “negative” control tissue samples were included in each IHC run, with the negative controls being represented by brain tissue sections of either Brucella spp.-infected striped dolphins from which the primary anti-B. melitensis antibody had been omitted, or by brain tissue slides of striped dolphins infected by other neurotropic pathogens like Cetacean Morbillivirus. All this information has now been properly included into the current ms revision, after having been preliminarly reported in the aforementioned validation study of the herein employed anti-B. melitensis Mab (Di Francesco et al., Vet. Ital., 2019).   

  1. Line 170. This IHQ has been tested for other gram-negative bacteria to rule out the cross-reactivity with other pathogens.

Reply: As previously mentioned, this information has now been included in the “Discussion” section of the present ms revision. However, no apparent cross-reactivities with other Gram-negative bacteria were found when performing IHC analyses with the anti-B. melitensis MAb utilized in this work, as previously reported in an ad hoc validation study published elsewhere (Di Francesco et al., Vet. Ital., 2019).

  1. Line 204-205 This sentence needs to be reworded as discussed above, there is not enough information within this study to elucidate this subjective conclusion. Other bacteria or conditions are not studied and other factors can cause this expression. No direct evidence of the role of B ceti in this expression in brain tissue is included (see comment No. 2).

Reply: As already mentioned, ad hoc laboratory investigations against viral (Cetacean Morbillivirus), protozoan (Toxoplasma gondii), bacterial (both aerobic and anaerobic) and fungal pathogens were also carried out on the herein investigated striped dolphin individuals. Furthermore, as already written before, we would like to thank this Reviewer (as well as Reviewer 3) for kindly suggesting us to cite/quote the interesting paper by Dr Erdogan (2013). In this respect, Brucella spp. entry into macrophages/histiocytes and the host’s response and other defense mechanisms against this microbial pathogen have not been clearly elucidated, thus far. More in detail, while PrPc mRNA silencing suppressed cell antioxidant systems, it also led to an upregulation of pro-inflammatory cytokines like IL-12 and TNF-alfa, with B. melitensis infection apparently exerting no effects on bacterial phagocytosis (Erdogan, 2013). Similar findings were reported in another study on macrophages infected with B. suis (Fontes et al., 2005), in contrast to the results of Watarai et al. (2003), who demonstrated that PrPc has relevant effects on phagocytosis and bacterial transport in B. abortus-infected macrophages.

In neurobrucellosis, however, the interaction mechanisms between the host cell and the corresponding Brucella spp. virulence factors that force the oxidative defense system of the host cells to react, together with the simultaneous PrPc action(s)/effect(s), remain largely unknown. Despite these unknowns, it is believed that PrPc, in addition to its antioxidant properties, could still play a role in Brucella spp. entry into cells through phagocytosis modulation/inhibition (Aydin et al., 2013), although we do not know yet if and to what extent PrPc mediates B. ceti entry into cetacean (striped dolphin) host’s macrophages/histiocytes.

All these thoughts and concepts have been duly included into the present ms revision, the conclusions of which have been carefully re-written, as wisely suggested by the Reviewer.

  1. Line 216-220. Neurobrucellosis is a chronic infection of bacterias of Brucella genera in the different infected species, no necessarily correlated with the level of inflammation or tissue damage, therefore the comparison between in vitro conditions and in vivo conditions of other gram-negative bacteria needs to be fully discussed with a number o possible variants affecting the expression of the infected animals versus higher expression in not infected animals.

Reply: These relevant issues (“pathogen-driven”, “host-driven”, or driven by both) have now been adequately dealt with in the present ms revision, as usefully suggested by the Reviewer.

  1. Line 244. This sentence needs to be reworded as discussed above, there is not enough information within this study to elucidate this subjective conclusion. Other bacteria or conditions are not studied and other factors can cause this expression. No direct evidence of the role of B ceti in this expression in brain tissue is included (see comment No.2) the comparison between in vitro conditions and in vivo conditions of other gram-negative bacteria needs to be fully discussed with a number of possible variants affecting the expression of the infected animals versus higher expression in not infected animals.

 Reply: We would like to thank this Reviewer, with whose opinions, comments and remarks we substantially agree. In this respect, we have taken into adequate account all of her/his valuable and precious comments, concerns and suggestions, thereby re-writing the concerned paragraphs of the ms “Discussion” section.

  1. Line 248-250. This conclusion is contrary to the information shown in Figure 2

Reply: As already stated, the variability seen in the PrPc expression patterns/levels in the brain tissue from the herein investigated striped dolphins could have a number of “explanatory keys”, with Brucella spp. infection and the intensity/magnitude of neurobrucellosis-associated/related microscopic changes being one of the possible options. Within this very complex and intricate context and scenario, we should also consider the role(s) played by individual host-related factors (including age, brain area/region sampled, physiological and/or simultaneously occurring pathological conditions, etc.), as well as by “extrinsic” factors such as concurrent infections caused by other bacterial and/or non-bacterial microorganisms, together with the effects of a huge number and different classes of “persistent environmental pollutants” (e.g. dioxins, PCBs, “flame retardants”, heavy metals, micro/nanoplastics). Indeed, we are dealing with free-ranging striped dolphins, which taxonomically belong to Odontocetes, an animals’ Sub-Order comprising “top predators” which do frequently accumulate and biomagnify (especially in the Mediterranean Sea basin), as such, heavy loads of the aforementioned environmental contaminants in their body tissues. In this respect, dioxins and dioxin-like compounds have been shown to exert pro-apoptotic effects on human neuronal cells (Morales-Hernández et al., 2012). Therefore, following dioxin-induced apoptosis, PrPC overexpression could be plausibly expected to occur on CNS (brain and spinal cord) host cell membranes, given the anti-apoptotic role of PrPC (Aguzzi et al., 2008).

Aguzzi A, Sigurdson C, Heikenwalder M. Molecular mechanisms of prion pathogenesis. Annu Rev Pathol (2008) 3:11–40. doi:10.1146/annurev.pathmechdis.3.121806.154326.

Morales-Hernández A, Sánchez-Martín FJ, Hortigón-Vinagre MP, Henao F, Merino JM. 2,3,7,8 Tetrachlorodibenzo-p-dioxin induces apoptosis by disruption of intracellular calcium homeostasis in human neuronal cell line SHSY5Y. Apoptosis (2012) 17:1170–81. doi:10.1007/s10495-012-0760-z.

As usefully suggested by the Reviewer, we have appropriately covered all these relevant issues in the “Discussion” section of our revised ms, which - we deem this important to be duly underlined - reports preliminary (and absolutely original) findings pertaining to PrPc expression in the brain tissue from stranded striped dolphins, either neurobrucellosis-affected (with a simultaneously occurring Brucella spp. infection) or not. Being these, to the best of our knowledge, the first “brain PrPc expression profile data” hitherto available in the biomedical literature for/in a Brucella spp.-susceptible cetacean species, we believe that, despite their intrinsic limitations, such data could serve as a reliable basis for future studies addressing the pathogenesis of neurobrucellosis in Brucella spp.-infected striped dolphins (as well as in other susceptible cetacean species), which is far from having been elucidated.   

  1. Line 250-260 See comment 2 This conclusion is contrary to the information shown in Figure 2

Reply: See comments provided above, please.

Reviewer 3 Report

This study reports the increased expression of cellular prion protein expression in the brain tissue from Brucella ceti-infected dolphins. The results of the study are clearly presented and discussed.

There are, however, several problems with the references in the introduction and discussion and the discussion could be improved.

In the introduction, references 8 and 9 are used to support the involvement of PrPc in Alzheimer's disease and in HIV infection. After consulting these articles, unless I am mistaken, these references do not seem to me to be appropriate. Similarly, references 14,15,16,17 are cited after the sentence "previous studies have demonstrated a role for PrPc in allowing B. abortus entry into macrophages from experimentally challenged mice". Many of these references are reviews and not experimental studies. Reference 17, for example, is a simple review that does not demonstrate a role of PrPc in B. abortus entry into macrophages. This profusion of references gives the reader the impression that numerous studies have documented the role of PrPc in infection by Brucella abortus, whereas this is not the case.

Several paragraphs in the discussion make general statements and do not cite any references. For example, there is no reference for : "The role of PrPc as a precursor of PrPSc during the onset of TSEs has been extensively investigated, with PrPc additionally playing an important role in T lymphocytes' and phagocytes' functioning, thereby acting as an immune response regulator under both physiological and pathological conditions. Furthermore, PrPc has been also identified as a potential pathogenetic factor and as a biomarker for other neurodegenerative disorders like Alzheimer's disease, as well as for bacterial and viral infections."

On the whole, the authors argue that it is well documented that PrPC is involved in Brucella infection, although this phenomenon has been little studied and remain controversial. This has only been shown in mice with Brucella abortus. As the mouse is not the natural host of Brucella abortus, the results obtained in this model should therefore be taken with caution. In this perspective, the article "Silencing of PrP C (prion protein) expression does not affect Brucella melitensis infection in human derived microglia cells" by Erdogan et al. (DOI: 10.1016/j.rvsc.2013.06.007) should be discussed. 

Author Response

Comments and Suggestions for Authors

This study reports the increased expression of cellular prion protein expression in the brain tissue from Brucella ceti-infected dolphins. The results of the study are clearly presented and discussed.

There are, however, several problems with the references in the introduction and discussion and the discussion could be improved.

In the introduction, references 8 and 9 are used to support the involvement of PrPc in Alzheimer's disease and in HIV infection. After consulting these articles, unless I am mistaken, these references do not seem to me to be appropriate. Similarly, references 14,15,16,17 are cited after the sentence "previous studies have demonstrated a role for PrPc in allowing B. abortus entry into macrophages from experimentally challenged mice". Many of these references are reviews and not experimental studies. Reference 17, for example, is a simple review that does not demonstrate a role of PrPc in B. abortus entry into macrophages. This profusion of references gives the reader the impression that numerous studies have documented the role of PrPc in infection by Brucella abortus, whereas this is not the case.

Reply: We have greatly appreciated the valuable and useful comments, remarks and suggestions kindly made by the Reviewer with reference to the bibliographic citations quoted in the ms. In this respect, we have repositioned reference 8 from where it originally was (being it an inappropriate citation there, as correctly adviced by the Reviewer), thereby quoting it alongside references 10 and 11. As far as reference 9 is concerned, we have checked again and we feel it could remain where it was, being it a review article on PrPc in health and disease (human prion diseases).  

Several paragraphs in the discussion make general statements and do not cite any references. For example, there is no reference for: "The role of PrPc as a precursor of PrPSc during the onset of TSEs has been extensively investigated, with PrPc additionally playing an important role in T lymphocytes' and phagocytes' functioning, thereby acting as an immune response regulator under both physiological and pathological conditions. Furthermore, PrPc has been also identified as a potential pathogenetic factor and as a biomarker for other neurodegenerative disorders like Alzheimer's disease, as well as for bacterial and viral infections."

Reply: We would like to thank the Reviewer for these highly valuable and precious comments and remarks on her/his behalf. In this respect, we have fixed the mistake/error made in the original ms, with special emphasis on the following sentence: Within lymphoid tissues, PrPc may also undergo an up- or downregulation during adaptive immune responses”. In fact, reference 8 was erroneously cited/quoted here instead of reference 28, as it has now been corrected in the present ms revision. The same also applies to the following sentence: Beside TSEs, PrPc is pathogenetically involved in other neurological disorders and infectious diseases, including Alzheimer's disease and human immunodeficiency virus (HIV) infection”, the correct bibliographic citation for which is not represented by reference 8 (as it happened in the original ms text/version), but rather by reference 28 (now correctly replacing the aforementioned bibliographic citation in the current ms revision).

As far as the other bibliographic references are concerned, while we should honestly admit that we had cited/quoted a (likely) excessive number of reviews in our original ms, it should be still emphasized that quite few studies reporting experimental data are available on the fascinating topic of “PrPc expression and biological significance in the pathogenesis of infectious diseases (other than prion diseases)”. Based upon the limited amount of experimental (and often conflicting) data available on this topic, coupled with the valuable and useful suggestions kindly provided by this Reviewer, we have decided to cite only two out of the three review articles previously quoted in the original ms, thereby deleting reference 16 (a book’s chapter) from the revised ms text/version.

On the whole, the authors argue that it is well documented that PrPC is involved in Brucella infection, although this phenomenon has been little studied and remain controversial. This has only been shown in mice with Brucella abortus. As the mouse is not the natural host of Brucella abortus, the results obtained in this model should therefore be taken with caution. In this perspective, the article "Silencing of PrP C (prion protein) expression does not affect Brucella melitensis infection in human derived microglia cells" by Erdogan et al. (DOI: 10.1016/j.rvsc.2013.06.007) should be discussed. 

Reply: This Reviewer is absolutely correct when stating that “As the mouse is not the natural host of Brucella abortus, the results obtained in this model should therefore be taken with caution.

While we have carefully addressed in the “Discussion” section of our revised ms these highly valuable and precious comments, remarks and suggestions kindly made by the Reviewer, we would also like to warmely thank her/him (as well as Reviewer 2) for kindly suggesting us to cite/quote the interesting paper by Dr Erdogan (2013). In this respect, Brucella spp. entry into macrophages/histiocytes and the host’s response and other defense mechanisms against this microbial pathogen have not been clearly elucidated, thus far. More in detail, while PrPc mRNA silencing suppressed cell antioxidant systems, it also led to an upregulation of pro-inflammatory cytokines like IL-12 and TNF-alfa, with B. melitensis infection apparently exerting no effects on bacterial phagocytosis (Erdogan, 2013). Similar findings were reported in another study on macrophages infected with B. suis (Fontes et al., 2005), in contrast to the results of Watarai et al. (2003), who demonstrated that PrPc has relevant effects on phagocytosis and bacterial transport in B. abortus-infected macrophages.

In neurobrucellosis, however, the interaction mechanisms between the host cell and the corresponding Brucella spp. virulence factors that force the oxidative defense system of the host cells to react, together with the simultaneous PrPc action(s)/effect(s), remain largely unknown. Despite these unknowns, it is believed that PrPc, in addition to its antioxidant properties, could still play a role in Brucella spp. entry into cells through phagocytosis modulation/inhibition (Aydin et al., 2013), although we do not know yet if and to what extent PrPc mediates B. ceti entry into cetacean (striped dolphin) host’s macrophages/histiocytes.

All these thoughts and concepts have been duly included into the present ms revision, the conclusions of which have been carefully re-written, as wisely suggested by this Reviewer.

Erdogan, S., Duzguner, V., Kucukgul, A., Aslantas, O. Silencing of PrPC (prion protein) expression does not affect Brucella melitensis infection in human derived microglia cells. Research in Veterinary Science 95 (2013) 368-373.

Fontes, P.; Alvarez-Martinez, M.T.; Gross, A.; Carnaud, C.; Kohler, S.; Liautard, J.P. Absence of Evidence for the Participation of the Macrophage Cellular Prion Protein in Infection with Brucella suis. Infection and Immunity, 2005, Vol. 73, No. 10, pages 6229–6236. doi:10.1128/IAI.73.10.6229-6236.2005.

Aydin, M.; Gilmore, D.F.; Erdogan, S.; Duzguner, V.; Ahn S. The Role of Cellular Prion Proteins (PrPC) on Microglial Brucella Infections. Agric. Food Anal. Bacteriol. 2013, Vol 3, pages 268-280.

Round 2

Reviewer 1 Report

Thank you for your revisions and comments. I believe the manuscript is much improved and provides valuable information to the readership. 

Author Response

Rebuttal Letter/Note

We have carefully revised our manuscript from the English language and style standpoints, as usefully suggested by this Reviewer, thereby correcting some minor errors/mistakes found throughout/across the entire text.

Again, thank you very much to this Reviewer (as well as to the other two Referees and to the Handling Editor) for the highly valuable, precious and constructive comments, remarks and suggestions kindly made on our manuscript, which have allowed us to greatly improve the overall quality of our work.

With our best wishes and our warmest regards.

Reviewer 2 Report

Angelucci et al, have applied most of the recommendations of the previous review.

Please consider the following points:

Line 62 Brucella ceti should be italicized

Line 63 Stenella coeruleoalba should be italicized

Line 64  B. ceti should be italicized

Line 65  B. ceti should be italicized

Line 67 B. abortus should be italicized

Line 72  B. ceti should be italicized

Line 77  B. ceti should be italicized

Line 78 Escherichia coli should be italicized

Line 79  B. ceti should be italicized

Line 80  Brucella ceti should be italicized

Line 81 Stenella coeruleoalba should be italicized

Line 83 B. ceti should be italicized

Line 84 B. ceti should be italicized

Line 86 B. ceti should be italicized

Line 90 B. ceti should be italicized

Line 94 B. ceti should be italicized

Line 187 B. ceti should be italicized

Line 189 B. ceti should be italicized

Line 190 Escherichia coli should be italicized

Line 191 B. ceti should be italicized

Line 193  Brucella should be italicized

Line 194 B. abortus should be italicized

Line 195 B. ceti should be italicized

Line 215-217 Other type font was used

Line 468-471 Other type font was used

Line 470 media should not be italicized

Line 564 uninfected should be changed by no microscopic evidence of CNS lesion. Since serology and culture were not performed in other organs or fluids the authors do not rule out the infection with brucellosis in other organs of these neurobrucellosis negative controls. If they did perform the serology and bacteriology from other tissues and samples this should be described in detail to confirm that these animals are non-brucellar animals. PCR is not the gold standard for the diagnosis of brucellosis.

Line 468-471 Other type font was used

Line 752-758  Other type font was used

Line 193 Brucella should be italicized

Line 849-862 Other type font was used

Line 999 and 1001-1002 repeat the same conclusion

Line 1002-1005 Other type font was used

Line 1224 B. ceti should be italicized

Line 1350 Brucella should be italicized

Line 1353 Brucella should be italicized

Line 1198 We don’t know yet sounds ambiguous and generates subjective conclusions please reword.